# Usual care and a self-management support programme versus usual care and a relaxation programme for people living with chronic headache disorders: a randomised controlled trial protocol (CHESS)

Shilpa Patel [1], Felix Achana,[2] Dawn Carnes [3], Sandra Eldridge,[3] David R Ellard [1], Frances Griffiths [4], Kirstie Haywood,[4] Siew Wan Hee [2], Dipesh Mistry,[1] Hema Mistry [5], Vivien P Nichols [1], Stavros Petrou [2], Tamar Pincus,[6] Rachel Potter [1], Harbinder Kaur Sandhu,[1] Kimberley Stewart,[1] Stephanie Taylor [3], Martin Underwood [1], Manjit Matharu[7]

For numbered affiliations see end of article.

**Correspondence to**
Dr Shilpa Patel;
shilpa.patel@warwick.ac.uk

## ABSTRACT

**Introduction** Chronic headaches are poorly diagnosed and managed and can be exacerbated by medication overuse. There is insufficient evidence on the non-pharmacological approaches to helping people living with chronic headaches.

**Methods and analysis** Chronic Headache Education and Self-management Study is a pragmatic randomised controlled trial to test the effectiveness and cost-effectiveness of a self-management education support programme on top of usual care for patients with chronic headaches against a control of usual care and relaxation. The intervention is a 2-day group course based on education, personal reflection and a cognitive behavioural approach, plus a nurse-led one-to-one consultation and follow-up over 8 weeks. We aim to recruit 689 participants (356 to the intervention arm and 333 to the control) from primary care and self-referral in London and the Midlands. The trial is powered to show a difference of 2.0 points on the Headache Impact Test, a patient-reported outcome measure at 12 months post randomisation. Secondary outcomes include health related quality of life, self-efficacy, social activation and engagement, anxiety and depression and healthcare utilisation. Outcomes are being measured at 4, 8 and 12 months. Cost-effectiveness will be expressed in terms of incremental cost per quality-adjusted life year gained.

**Ethics and dissemination** This trial will provide data on effectiveness and cost-effectiveness of a self-management support programme for chronic headaches. The results will inform commissioning of services and clinical practice. North West – Greater Manchester East Research Ethics Committee have approved the trial. The current protocol version is 3.6 date 7 March 2019.

**Trial registration number** ISRCTN79708100.

## Strengths and limitations of this study

► This trial presents a high-quality randomised controlled trial designed to determine the effectiveness and cost-effectiveness of a group self-management support programme for adults with chronic headaches.

► The intervention underpinning the trial is a complex intervention that has been designed using the best available evidence and theory. The full details of the intervention development and design have been peer reviewed and published.

► A separate mixed-methods process evaluation is running alongside the trial and will explore fidelity of the intervention and implementation. A detailed process evaluation protocol has been peer reviewed and published.

► The success of the trial will be dependent on the ability to deliver this complex group intervention, ensure fidelity and subsequently obtain follow-up data in this population.

## INTRODUCTION

Headache disorders are a major cause of pain and disability.[1] A definition of a chronic headache disorder used in epidemiological studies is that the person has headache for 15 or more days per month for at least 3 months.[2–4] Other authors prefer the term chronic daily headache, but this is less clearly defined.[5] Chronic headache disorders mainly affect the young adult population, many of whom have both work and family commitments.[6] The most common chronic headache disorders are chronic tension type headache, chronic

migraine and medication overuse headaches (MOHs).[1] Tension type headache and migraine are primary headaches. Medication overuse is a secondary headache that can develop in people with frequent acute headaches who take analgesics or specific antimigraine compounds (eg, triptans) for ≥10 days per month.

Around 2%–4% of the global population experience chronic daily headaches.[7] Between a quarter and half of those affected also have MOH, which has a prevalence of 1%.[8–10] Many people with chronic headache have undiagnosed chronic migraine.[11] More appropriate use of pharmacological treatment has the potential to improve outcomes for people living with chronic headache disorders. This might mean introducing migraine prophylactic drugs or using fewer analgesics/triptans. Depression, anxiety, poor sleep, stress and poor self-efficacy for managing headaches are prognostic markers for a poor outcome in chronic headache disorders.[12] Limited qualitative data indicate that chronic headache disorders can directly and indirectly drive behaviour: that people live with the 'spectre of headache' (an array of concerns people with chronic headaches have to take into account when forward planning) and that headaches lead to strained relationships.[13] For people living with chronic migraine or chronic tension type headache, non-pharmacological self-management approaches may improve headache-related symptoms while not affecting headache frequency.[14] Group interventions using a cognitive behavioural approach, mindfulness and educational components may be more effective than alternative interventions.[14]

Supportive self-management approaches are well established in the management of several chronic painful disorders, but this is not the case for chronic headache disorders.[15–17] The National Institute for Health and Care Excellence (NICE) guidance in England, for example, only makes one positive recommendation for a non-pharmacological treatment for people living with chronic migraine or chronic tension type headache: to consider a course of acupuncture for people with chronic migraine or chronic tension type headache.[18]

We describe here the protocol for a randomised controlled trial (RCT) to estimate the effectiveness and cost-effectiveness of a group self-management support programme for adults with chronic headaches arising from migraine or tension type headache, with or without medication overuse. The Chronic Headache Education and Self-management Study (CHESS) intervention consists of a group education and self-management support programme plus a tailored one-to-one headache consultation exploring medication optimisation, lifestyle factors and goals. This active arm of the trial is being compared with usual care plus a relaxation programme.

## METHODS AND ANALYSIS

The methods section is structured in accordance with the SPIRIT 2013 recommendations.[19] Here we provide an overview of the trial plus a brief description of the proposed health economic evaluation, process evaluation and patient and public involvement (PPI).

The primary objective is to estimate the clinical and cost-effectiveness of a group education and self-management programme for people living with chronic headache recruited from primary care when compared with a general practitioner (GP) care plus relaxation control group.

### Participants, interventions and outcomes
#### Study setting
Primary care settings in two localities: London and the Midlands.

#### Eligibility criteria
The population of interest are adults living with chronic migraine or chronic tension type headache, with or without medication overuse. Part of the challenge in interpreting the findings of research in people living with chronic headaches is the often poor reporting of participants' phenotypic characteristics, meaning it is not possible to draw conclusions for specific chronic headache types.[13 20] For this trial, it is important that we are able to define our participants' headache type, or types, accurately and exclude those with non-eligible headache types. We, therefore, developed and validated a logic model for use in a telephone interview by a nurse (non-headache expert) that would allow us to identify and classify people meeting our entry criteria while also excluding people with secondary headaches, other than medication overuse and other causes of primary headaches (table 1).[21]

Pregnancy is not an exclusion criterion. However, any pregnant women randomised to the active intervention will be advised to speak to their GP with regards to medication, and the CHESS intervention nurses will not discuss this with them during the consultation for safety reasons.

Our previous experience is that the challenges of running non-English language group interventions for chronic painful disorders are too great to do successfully within an RCT.[22] Furthermore, the main outcomes are not validated in those languages other than English, which are relevant to a UK context. For these reasons, we are excluding people who are not fluent in written and spoken English.

In this pragmatic trial, our entry criteria reflect the point in the care pathway where our intervention will be offered. Specifically, general practitioners will refer people they identify with chronic headaches into the service. Thus, there is not the pre randomisation run-in one might expect in a more explanatory drug trial, and we base assessment of the presence of chronic headache on a single telephone assessment and include a mixture of headache types.

#### Interventions
The intervention consists of a group education and self-management programme, an 8-week headache paper

**Table 1** Inclusion and exclusion criteria

| Inclusion criteria | Exclusion criteria |
| --- | --- |
| 1. Able and willing to comply with the study procedures and provide written informed consent.<br>2. Aged ≥18 years (no upper limit).<br>3. Living with chronic headache; defined as headache on 15 or more days per month for at least the preceding 3 months.<br>4. The nurse telephone classification interview confirms headache type to be chronic migraine, or chronic tension type headache, with or without medication overuse headache.<br>5. Fluent in written and spoken English. | 1. Unable to attend the group sessions.<br>2. No access to a telephone (for classification interview).<br>3. Has an underlying serious psychological disorder with ongoing symptoms that preclude or significantly interfere with participation in the group intervention.<br>4. Previous entry or randomisation in the present trial.<br>5. Currently participating in another clinical trial of headache treatments or unregistered medicinal product or less than 90 days have passed since completing participation in such a trial. |

diary, a one-to-one nurse-led consultation and follow-up telephone calls, where necessary, for up to 8 weeks. We have described its development in detail elsewhere.[23] We summarise it briefly here.

The programme is run over 2 days in a 2-week period for around 10 participants per group (target 6–12 participants) and is facilitated by two intervention trained healthcare professionals, at least one of whom is a nurse (the second facilitator could be a nurse or other registered allied health professional such as psychologist, physiotherapist, chiropractor and occupational therapist). We originally planned to run sessions using a nurse and a lay facilitator living with chronic headaches. However, in our feasibility study, we found it difficult to recruit people living with chronic headaches to act as facilitators and of those that were recruited several found it difficult to commit to facilitation of courses because of the unpredictability of their condition. The intervention builds on a previously developed and tested educational and cognitive behavioural self-management intervention for people with chronic musculoskeletal pain.[24]

The aim of the course is to encourage and enable those with chronic headaches to recognise unhelpful thought patterns and behaviours that contribute to their headache burden and to do something about them. The course provides participants with an overall toolbox of strategies that could help in the management of their headaches. These strategies include psychological techniques to change perceptions and feelings about living with chronic headaches as well as more practical strategies around lifestyle factors and medication. The group intervention is delivered using a range of methods including: group discussions, ideas generation, sharing narratives and experiences, problem solving, role play and taster activity sessions. The programme includes a range of behaviour change techniques including: barrier identification, general encouragement, instruction from the group facilitators, provision of feedback and allowing opportunities for social comparison in the group. We also include an educational video (available on disc and online) following feedback from our PPI group on the importance of having something to show family and friends about their 'invisible' disorder.

The sessions take place on weekdays and, where possible, during school hours to accommodate those with child care responsibilities. Sessions are held in easily accessible community venues and GP practices.

Participants in the intervention arm are asked to complete a paper headache diary for a period of up to 8 weeks prior to the 2-day programme. A one-to-one, individually tailored nurse-led consultation follows the 2-day programme. During this session, the nurse classifies the participant's headache type, discuss medication and lifestyle factors and explore participants' goals. This discussion is backed up by written information (for patient and GP), consistent with NICE guidance, to support shared informed decision making between the patient and their GP, about medication choices.[25] All participants will be offered telephone follow-up for up to 8 weeks. The frequency of these follow-up calls will be individually negotiated and agreed with participants during the one-to-one consultation. Full details of the intervention content have been published elsewhere.[23]

### Control intervention

Previous studies of this nature have reported that a usual care control arm was not an incentive to join the study.[15] We know from experience that patients enjoy the relaxation part of pain self-management programmes.[17] Therefore, as an incentive to participate, the control group receive standard usual care plus a relaxation CD and instructions for use over the duration of the study, up until final follow-up at 12 months. Participants are free to continue using the relaxation CD thereafter should they wish.

As all participants in the trial have a headache classification interview prior to randomisation, we feed back the results of that classification interview to control participants, and their GPs, together with advice on headache management. In this way, we ensure all participants receive best usual care.

The relaxation CD or downloadable MPG file (from the CHESS website) consists of a 17 min relaxation audio script that starts by focusing on the breath before talking the participant through progressive relaxation of muscles. The participants in the control group choose

when, where and how often they do the relaxation but are advised to try it two to three times a week or as often as they feel appropriate over the course of the study.

## Facilitator recruitment and training

All group sessions have two facilitators: one of whom must be a registered nurse, while the other facilitator is any allied healthcare professional, registered with a regulatory body, with an interest in this patient population and condition. Facilitators were recruited via advertisements in allied healthcare profession organisations and trained over a 2-day period to deliver the self-management intervention. Nurses received a further day of training to cover the one-to-one consultations. The training course covers the content of the intervention, facilitation skills, managing groups and trial procedures. Those undergoing the training were assessed using a learning assessment form at the end of the training course to check their knowledge and understanding. Those evaluated to be competent were asked to deliver the intervention. Those who either struggled with some elements of the content or expressed any concerns about delivering the intervention were provided with additional one to one support by the trainers.

## Intervention quality assessment

Quality assurance of the intervention is important to ensure that the trial intervention is delivered consistently and in accordance with the trial protocol and manuals. As part of the intervention development, we have produced two very detailed manuals: the first that covers the 2 days of group sessions and the second that details the process for the one to one consultations. All facilitators have been appropriately trained on the delivery of the intervention in accordance with the manuals. They have been instructed to use the manuals to guide them through delivery of the intervention.

To assure the quality of the course delivery, we will aim to observe each facilitator. We will observe each facilitator early in their facilitation to allow any difficulties or challenges to be addressed. Subsequently, they will be observed midway through. Observations will be by session, and the number of sessions observed will depend on the ability of the observer to capture the required information.

The observations will be conducted by members of the CHESS team who have knowledge of the intervention and its delivery. Observers will complete an observation form that will address facilitator skills and adherence to content delivery. Feedback will be provided on the day where possible. If this is not possible, the observer will arrange to contact the facilitator by phone. The study team will discuss any difficulties with the facilitator to minimise impact on the rest of the course and to help with the delivery of future courses. For any facilitators struggling, the central research team will monitor their personal reflections for the remaining duration of the course and follow-up with phone calls if required.

## Outcome measures: primary outcome measure; headache-related quality of life

Our primary outcomes is headache-related quality of life assessed at the primary endpoint, 12 months after randomisation. For sample size determination, we have specified the Headache Impact Test (HIT-6), a six-item patient-reported outcome measure (PROM), as our measure of the primary outcome.[26] The HIT-6 provides a short overall assessment of headache impact – with items assessing fatigue, pain, social functioning, emotional well-being and cognition. Prior to selecting our outcome measures, we did a systematic review of PROMs for headaches.[27] Only for the HIT-6 did we find acceptable evidence supporting its use for our target population who may not have been given a headache diagnosis prior to study entry.

Another measure, shortlisted in the review, was the 14-item Migraine-Specific Quality of Life Questionnaire (MSQ v2.1) assesses the role restrictions, limitations and emotional impact of migraine.[27] The MSQv2.1 has acceptable evidence of psychometric properties following completion in a migraine population.[27–30]

To inform our selection of outcome measures, we undertook interviews (n=14) with people with chronic headache. Our participants described greater perceived relevance of the MSQ (v2.1) compared with the HIT-6. With permission from GSK, the copyright holders, we modified the MSQ (v2.1), changing the focus from 'migraine' to 'headache'; the modified measure was renamed as the 'Chronic Headache Quality of Life Questionnaire' (CHQLQ v1.0). In our ongoing work, we are assessing the performance of this modified instrument. In the event its performance is superior to the HIT-6, we will consider changing the primary outcome.

Both the HIT-6 and the CHQLQ v1.0 will be collected via postal questionnaire at baseline, 4, 8 and 12 months post randomisation.

## Outcome measures: other secondary outcomes measures

The following secondary outcome measures are being collected:

1. *Headache days:* our main outcome describing headaches days will be headaches days in the preceding month reported at baseline and follow-up postal questionnaires. We will also estimate total headaches days over the whole study period from patient-reported data collected via a smartphone app.
2. *Headache impact:* we will assess headache impact based on their typical duration and severity reported in baseline and follow-up questionnaires.
3. *Composite headache outcome:* we will produce a composite headache outcome of headache days × severity × duration using data from the smartphone app and questionnaires.
4. *Generic health-related quality of life:* we have included two standard measures of health-related quality of life—the (SF-12 V2) Short Form Survey 12 Version 2 and (EQ-5D-5L) EuroQol five dimension scale.[31–33] There

is limited, but acceptable, evidence supporting the use of the SF-12 V2 to assess overall quality of life in a headache population.[27] There is no such evidence for the EQ-5D-5L.[27] We will, therefore, use EQ-5D-5L primarily for our health economic analyses.[32]

5. *Emotional well-being:* Hospital Anxiety and Depression Scale (HADs) – psychological distress is extremely common in people living with chronic pain. HADs has been used in many previous studies of chronic pain, including the COPERS (Coping with persistent Pain, Effectiveness Research into Self-management) study where we identified positive effects on both anxiety and depression in a chronic pain population.[22 34]

6. *Self-efficacy:* Pain Self-Efficacy Questionnaire (PSEQ) – self-efficacy is an important mediator for how self-management interventions may improve patient outcomes. It is important, therefore, to measure change in self-efficacy as part of understanding the causal pathway for any change and informing our process evaluation.[35] We have previously reviewed measures of self-efficacy and concluded that PSEQ is the most appropriate choice for studies of this nature; although all current measures have limitations.[36]

7. *Social activity: Social Integration Subscale (SIS) of the Health Education Impact Questionnaire (heiQ)* – chronic headache can result in a disrupted lifestyle and a reduced quality of life both during and between attacks; the impact of chronic headache on an individual's ability to commit to social plans is an important aspect of quality of life. Successful treatment should seek to improve both overall quality of life, as well as an individual's quality of life during the attack, including their ability to integrate in society. A well-developed, condition-specific measure should seek to capture these distinctions. The five-item SIS is one of eight domains contained within the heiQ, a measure of the impact of patient education programmes in chronic conditions.[37]

8. *Bodily pain:* chronic headache commonly coexists with other chronic painful disorders such as low back pain.[38–41] The CHESS intervention might affect the troublesomeness of other bodily pains. We will collect these data using a previously validated Troublesomeness Grid.[42]

At baseline we also collect data on age, gender, ethnic group, age at leaving full-time education and current work status.

We will collect follow-up data at 4, 8 and 12 months after randomisation by postal questionnaire survey. A £5 high street voucher is enclosed as a token of our appreciation at each initial time point. We will send out two postal follow-up reminders. In the event that no response is obtained, we will aim to collect our primary clinical outcome data by telephone. This includes the HIT-6 and EQ-5D-5L.

The smartphone data will be collected weekly for 6 months from initial eligibility and then monthly until 12 months after randomisation. A paper version of the app will be available to those who do not use a smartphone.

If there are missing data (for our key clinical outcomes), this will be followed up with the participant who completed the form, as soon as possible by a member of the team over the phone. We will phone the participant and enter the correct information onto the form; this will be initialled and dated.

### Participant timelines
Figure 1 shows the recruitment flow, and table 2 shows the study timelines.

### Sample size
For the purposes of our sample size calculation, the primary clinical outcome is the difference in mean HIT-6 score at 12 months post randomisation between the self-management group programme and the usual care relaxation therapy (control arm). The HIT-6 outcome measure is a continuous scale with higher values indicating more severe impact on daily life. From our systematic reviews, we anticipate a worthwhile difference to be 2.0, that is, mean outcome in the control arm is 2.0 units higher than for the intervention.[27 43] In our feasibility study (n=114), the SD of the HIT-6 score at baseline was 6.87.[44]

Participants are randomised to either the self-management group or usual care and relaxation therapy. In this design, there may be a clustering effect in the self-management group and not in the control arm, which needs to be allowed for in the sample size calculation. Based on similar trials, we assume that the intraclass correlation coefficient is 0.01.[22] The anticipated average size of the self-management programme is 10.

The minimum sample size required was estimated using Moerbeek and Wong's method to account for grouping in one arm.[45] To detect a between-group difference of 2.0 with SD of 6.9 (a standardised mean difference of 0.29) and assuming that the ratio of the total variance in the self-management group to the relaxation therapy is 1 at the two-sided 5% significance level and at least 90% power, the sample size required is 523 participants (253 in the relaxation group and 270 in the self-management group).

In the feasibility study, the overwhelming majority of those recruited, approximately 95%, chronic migraine; just 5% had chronic tension type headache. We want to be able to draw definite conclusions on the specific subgroup of chronic migraine. Therefore, we will base our sample size and primary clinical outcome on the population with chronic migraine. Therefore, based on 95% of our sampled population with chronic migraine and accounting for a 20% loss to follow-up as above, the sample size we would require is 689 with 333 to the relaxation arm and 356 to the self-management programme.

In the feasibility study, we recruited around 1/1000 of practice population to take part[44]; we therefore anticipate we need to recruit from around 100 general practices with a combined list size of around 700 000.

In consultation with the Data Monitoring and Ethics Committee (DMEC), we will review the sample size

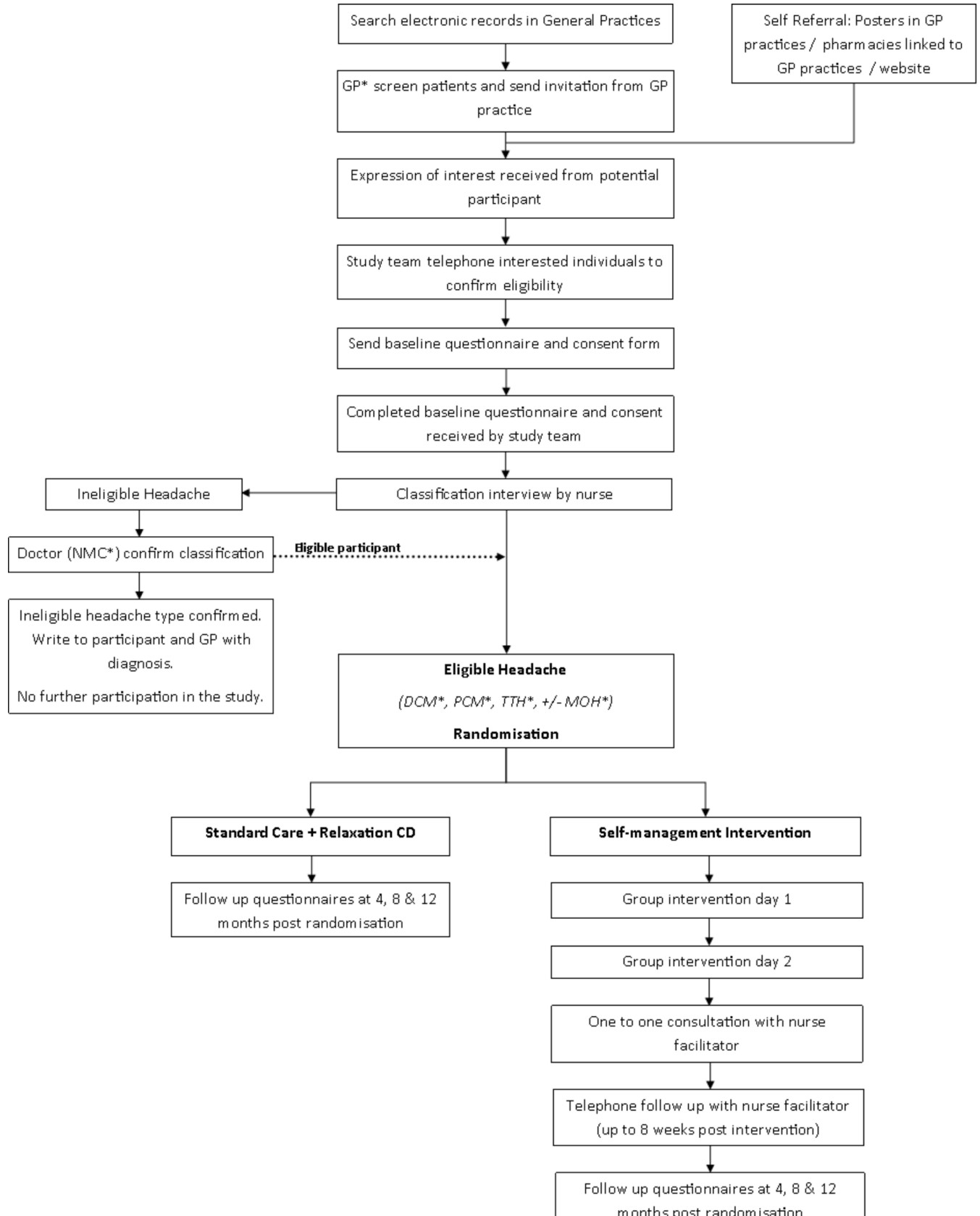

**Figure 1** This shows the recruitment flow chart.

*Genetal practitioner; National Migraine Center; Definite Chronic Migraine; Probable Chronic Migraine; Tension Type He adache; Medication Overuse He adache.

DMC, definite chronic migraine; GP, general practitioner; MOH, medication overuse headache; PCM, probable chronic migraine; TTH, tension type headache.

**Table 2** Schedule of enrolment, interventions and assessments

| Activity | Pre-enrolment to allocation | Randomisation to 12 weeks | 4 months | 8 months | 12 months |
|---|---|---|---|---|---|
| Expression of interest | × | | | | |
| Trial information | × | | | | |
| Download app to record headache patterns (weekly for 6 months and then monthly until the end of follow-up at 12 months) | × | × | × | × | × |
| Questionnaire demographic information, HIT-6, CHQLQ, headache days, SF12, EQ-5D-5L, HADs, PSEQ, HeiQ and troublesomeness | × | | | | |
| Consent | × | | | | |
| Headache classification interview | × | | | | |
| Allocation | | × | | | |
| Intervention plus completion of paper headache diary for duration of up to 8 weeks before attending group | | × | | | |
| Control | | × | | | |
| HIT-6, CHQLQ, headache days, SF12, EQ-5D-5L, HADs, PSEQ, HeiQ and troublesomeness | | | × | | |
| Two postal follow-up reminders and phone call for PO if needed | | | × | | |
| HIT-6, CHQLQ, headache days, SF12, EQ-5D-5L, HADs, PSEQ, HeiQ and troublesomeness | | | | × | |
| Two postal follow-up reminders and phone call for PO if needed | | | | × | |
| HIT-6, CHQLQ, headache days, SF12, EQ-5D-5L, HADs, PSEQ, HeiQ and troublesomeness | | | | | × |
| Two postal follow-up reminders and phone call for PO if needed | | | | | × |

CHQLQ, Chronic Headache Quality of Life Questionnaire; EQ-5D-5L, EuroQoL; HADs, Hospital Anxiety and Depression Scale; HeiQ, Health Education Impact Questionnaire; HIT-6, Headache Impact Test; PO, Primary outcome; PSEQ, Pain Self-Efficacy Questionnaire; SF12, Short Form 12-item Health Survey.

around halfway through recruitment to ensure we have recruited sufficient participants with chronic migraine and revise our estimates using within trial data on the variance of our primary outcome at baseline. This review will be based on the headache classification and actual baseline SD of our sampled population. We might also need to recruit some additional participants to ensure that the final group sessions at each site are adequately populated.

### Recruitment

Participants are identified and invited into the study in two ways. First, practices run electronic searches on their databases to identify people who have consulted with headaches or have been prescribed migraine specific drugs (eg, triptans and pizotifen) in the preceding 2 years. Practices then screen the lists for those it would be inappropriate to approach (eg, poorly controlled serious mental illness, terminal illness or known secondary causes of headache such as primary or secondary brain tumours) and send approach letters to the remainder.

People can self-refer to the trial. Participating general practices, the principal pharmacies used by their patients, are supplied with a study poster for display in patient areas inviting people to contact the study team if they have headaches and are interested in participating. Information about the trial is made available on the poster (in general practices and pharmacies) and websites (https://warwick.ac.uk/fac/med/research/ctu/trials/other/chess/ and https://warwick.ac.uk/fac/med/research/ctu/trials/chess. People who find about the trial through the internet or following media exposure, and can travel to sessions, can also self-refer to the trial.

Using both approaches will allow people receiving GP treatment for chronic headaches who are not coded in the GP system as having headaches, and those who are self-medicating their headaches, the opportunity to join the study. We anticipate that we will primarily recruit people registered with participating practices; however, we will not restrict recruitment to those registered with participating practices.

In addition to these two main recruitment strategies, a study press release was submitted in January 2018, which was picked up by various media outlets and generated further interest in the study from potential participants.

The study coordinating team contact people expressing an interest in the study and check that they are eligible, explain the study and obtain participant's verbal consent to start completing an electronic headache symptom severity, duration and frequency diary (or paper version where there is no access to the internet). The electronic diary is completed weekly for the first 6 months and subsequently monthly until the end of follow-up at 12 months. The electronic diary is used to identify any early effects of rebound headache in those with MOH. At this time participants are sent a baseline questionnaire and consent form to return in a freepost envelope.

Following receipt of baseline questionnaire and consent form, participants are contacted by phone for a nurse headache classification interview. People whose headache is confirmed to meet our entry criteria are then eligible for randomisation. If the nurse has a concern that the potential participant may have another headache type, there is a second interview with a doctor skilled in the management of headache disorders from the National Migraine Centre (http://www.nationalmigrainecentre.org.uk/). This is to ensure no one is inappropriately excluded and that people with a headache type needing more specific treatment are directed towards appropriate treatment. For example, 2/108 people in our feasibility study had cluster headaches.[44] In such cases, we write to the GP and the participant with details on the excluded headache type and explain that they no longer fit the inclusion criteria for the CHESS trial.

## Assignment of interventions
### Randomisation
The randomisation is stratified by geographical locality (Midlands and Greater London) and headache type (six possible headache types: chronic tension type headache, probable chronic migraine and definite chronic migraine with or without MOH) using minimisation. Randomisation takes place using an online application specifically developed for the CHESS Study by the Warwick Clinical Trials Unit (WCTU) programming team.

Randomisation takes place when there are around 20 participants who would be able to attend a local group if randomised to the active intervention. Typically these will come from 4 to 5 proximate general practices. However, people from other localities who are unable to attend a more local group and people who have self-referred to the study who are willing and able to travel to the sessions will also be included. By randomising in groups around 2 weeks prior to the start of a group, we are able to minimise any delay between randomisation and start of the intervention and ensure everyone has adequate follow-up from when they might have been exposed to the intervention. For those that are subsequently unable to attend

after randomisation to a group, we will offer them another group where possible.

### Allocation concealment and protection from bias
Allocation concealment will be maintained by using Warwick CTU's centralised randomisation service. All baseline data will be collected prior to randomisation. Blinding will be impossible for participants, facilitators and the central co-ordinating CHESS team. All data entry and checking is done blind to treatment allocation. The unbalanced randomisation means it is not possible blind the trial statistician to treatment allocation.

Our primary outcome is a participant completed outcome. Inevitably, participants will be aware of their treatment allocation. We will develop and sign off a detailed prespecified statistical analysis plan before allocation codes are released.

## Data collection, management and analysis
### Data collection methods
Data will be collected using two methods: postal survey questionnaires and a smartphone application (App) for headaches. All participants will be asked to complete a smartphone app about their headaches. If they do not have access to a smartphone or do not wish to use the app, a paper copy will be provided. Participants will initially complete the app weekly for up to 6 months to cover any period of withdrawal from medication, then monthly thereafter (still requiring them to reflect over the previous 7 days) until the end of the study at 12 months after randomisation. We are working with Clinvivo Ltd, a University of Warwick spin-out company specialising in electronic data collection, to capture data on headache frequency, duration and severity electronically using a smartphone app (table 3 shows the questions completed by participants and mode of response required). The data from the questions in the electronic diary will be numerical and downloaded into a WCTU database using a randomly generated 256-bit Advanced Encryption Standard (AES) key. At the end of the participants' 12 months, they will be provided with a summary report of their headache application data.

Reminders are sent to those participants who fail to downloaded the app or have not responded for more than 3 weeks. Reminders are sent by email or by post if there is no email address given.

| Table 3 Questions completed on the smartphone app | |
|---|---|
| **Question** | **Mode of response** |
| On how many of the last 7 days have you had a headache? | Insert number of headache days |
| On those days you had a headache, on average how long did they last? | Insert number of hours |
| On those days you had a headache, on average how severe were they? | Scale of 0–10 (with 0 being no pain and 10 extremely severe pain) |

A daily detailed paper headache diary is kept only by the intervention arm participants, and the data will not be analysed by the study team but kept as a record by the participants and used during the one to one consultation where it will be collected and kept by the research team.

## Data management

All questionnaire data received by the trial team are reviewed for completeness and entered onto a secure, backed-up bespoke database held at WCTU, which will be accessible only by authorised members of the team. Data from the smartphone app are downloaded securely (as described above) into the trial database. All data are checked when received, and any queries relating to the HIT6 CHQLQ or EQ5D are raised with the participant by telephone.

## Statistical analysis of effectiveness and harms

Our primary endpoint analyses will be based on the 12-month data.

Participants' characteristics and reported outcomes will be summarised as means and SDs (for continuous data) or frequencies and percentages (for categorical data) by treatment arms. Difference between baseline and the three follow-up time points (4, 8 and 12 months post randomisation) will be computed for the primary and secondary outcomes by treatment arms.

The primary analysis approach will be intention to treat; that is, the data will be analysed according to the treatment the participant was originally allocated to, irrespective of what they actually received. We will explore the possibility of carrying out a complier averaged causal effect analysis as a sensitivity analysis. Our primary analysis will be the difference between the self-management therapy (intervention) and the relaxation therapy (control) groups with a 95% CI in the population with chronic migraine – if the proportion of participants with chronic tension type headache is ≤15%. The hypothesis testing of the primary outcome will be two sided at the 5% level, and the main analysis will estimate the treatment effect using a multilevel model to account for clustering (the model used to design this main trial). We will also present results for the whole population (all headache types). If the proportion of chronic tension type headache is >15%, then the primary analysis will be according to the whole population of chronic headache (chronic migraine and tension type headache). Our experience is that NICE was specifically interested in data on specific headache types, excluding data that reported mixed population of people with chronic headaches. We will, therefore, in addition to our primary analyses, present the results (mean difference and 95% CI) for each of the two headache types with or without MOH separately and present results for those with or without medication overuse separately to facilitate future meta-analyses and inform future condition specific guidelines. All analyses will be adjusted for the randomisation stratification factors (types of headache and geographical locality), sex, age and the baseline value of the dependent variable.

Similar analyses will be performed for all the other secondary outcomes. Prespecified subgroup analyses using formal statistical tests for interaction will examine whether baseline anxiety, depression and severity are moderators of treatment effect.[46] We will assess the level of missingness in the primary outcome, and if required, we will use appropriate multiple imputation techniques to impute data and estimate the treatment effect as a form of sensitivity analysis. A full statistical analysis plan, including all primary and secondary analyses, will be written and signed off prior to releasing allocation codes to analysts and others involved in developing the plan.

Using the data from the smartphone app, we will generate a composite score for headache impact over the 1 year of follow-up as the function of headaches days × average duration × average severity. Presenting these data graphically will allow any early benefits or harms from the intervention to be identified.

## Health economic evaluation

Our economic evaluation will be conducted alongside the trial, and we will initially adopt a 1 year time horizon from both an NHS and personal social services perspective and, separately for the purpose of a sensitivity analysis a broader societal perspective, to estimate the cost-utility of the intervention. Resource use data will be collected to explore the costs of the delivery of the intervention and to estimate the key cost drivers. This will mainly consist of visits to the GP practice, medication usage and any adverse events (AEs) or length of stay in the hospital. In terms of costs to society, we will estimate time off work and any productivity losses associated with chronic headaches. Resource use information will be collected using self-completed postal questionnaires completed at 4, 8 and 12 months after randomisation, as well as the use of routine health service data collected from general practice records. Resource input will be valued using national estimates of unit costs such as the Prescription Cost Analysis database or the Unit Costs of Health and Social Care compendium.[47] Preference-based health-related quality of life outcomes will primarily be assessed through the completion of the EQ-5D-5L at each follow-up point.[48] Quality-adjusted life-years (QALYs) will be calculated as the area under the baseline-adjusted utility curve and will be calculated using linear interpolation between baseline and follow-up utility scores.

The results of the economic evaluation will be presented using incremental cost-effectiveness ratios, expressed in terms of incremental cost per QALY gained, and cost-effectiveness acceptability curves generated via non-parametric bootstrapping.

More extensive economic modelling using decision analytic methods will extend the target population, the time horizon to 5 years as the long-term natural history is unclear and the decision context, drawing on best available information from the literature together with

stakeholder consultations to supplement the trial data. Longer term costs and consequences will be discounted to present values using nationally recommended discount rates recommended for health technology appraisal. We will use probabilistic sensitivity analysis to estimate the impact of uncertainty over model parameters. We will also use simple sensitivity analysis to assess the robustness of the results to changes in deterministic parameters such as medication dosages, costs, discount rate and time horizon for patients presenting with chronic headaches. We will also explore cost-effectiveness of the intervention by conducting subgroup analyses for the different headache types.

## Monitoring, ethics and dissemination

### Programme steering committee (PSC)

We have established a PSC to oversee the whole programme of work. The PSC functions specifically as a trial steering committee specifically for this trial. The PSC consists of seven members including two patient representative. The PSC will be responsible for oversight and monitoring and will consider any information from surveillance activity of other research in the field.

### Data monitoring and ethics committee

A DMEC has been convened consisting of an independent statistician, a triallist, a clinician and a lay member. Confidential reports that summarise the trial data and safety data will be reviewed by the Data Monitoring Committee (DMC). The DMC will advise the PSC as to whether to continue, amend or terminate the trial based primarily on safety and efficacy considerations

### Harms/AEs

An AE or serious AE is any event that takes place on the way to, during or on the way home from the intervention course. This includes the 2 days of the group course and the one-to-one nurse appointment. Our experience across multiple studies of group interventions is that AEs that are directly attributable to the intervention are rare.

Events during the session, such as severe psychological disturbance, any mild or moderate levels of emotional distress as a result of discussing experiences of living with chronic headache, a fall during travel to and from the venue or any other AEs, occurring during the delivery of the intervention will be reported to the trial co-coordinator by the intervention team. Any short-term increase in headaches as a consequence of medication withdrawal will be captured using the smartphone app (or paper version for those without access to a smartphone).

All AEs will be managed in line with Warwick CTU's standard operating procedures.

### Process evaluation

A separate mixed-methods process evaluation is running alongside the trial. This will explore both the process of implementation the trial and the process of delivering and receiving the intervention. The process evaluation protocol is reported elsewhere.[49]

The fidelity of the delivery of the intervention will be assessed by this team in three ways, through: audio recording of the groups, facilitator reflection and participant feedback. This will assess facilitator competence, adherence to the course manual and trial procedures and participant interaction and engagement.

### Patient and public involvement

We have had substantial PPI in the feasibility study prior to finalising this protocol. Lay members were involved in the development of the classification interview, development of the intervention, PROM selection and conduct and management of the study via the independent programme steering and trial management group.

Our trial management group includes our lay coapplicants who are representatives of three leading UK migraine charities (The Migraine Trust, Migraine Action (before their merger with Migraine Trust) and National Migraine Centre).

We have developed a lay steering group who are and will be collaboratively involved during the study. At key points in the programme we will approach the lay steering group for input, this will include at the end of the trial when results are disseminated.

## ETHICS AND DISSEMINATION

The University of Warwick (Research Impact Services, University of Warwick, Coventry CV4 7AL) is the sponsor for the study.

The trial is conducted in accordance with the principles outlined in the Declaration of Helsinki and conforms to Good Clinical Practice (GCP) and Standard Operating Procedures (SOPs) set-out by the WCTU. As reported above, we have convened a PSC and DMC to oversee the trial.

All identifiable data are anonymised and treated as confidential. Participants are informed that they are free to withdraw at any time during any phase of the work.

The findings from this trial will be disseminated via a final report to the National Institute for Health Research (NIHR), presentations at conferences and publications in high-quality peer-reviewed journals.

For the healthcare professionals involved in the study, we will disseminate results of the study through the study website. We will also host a meeting to present the trial results to commissioners and clinicians. For the participants, we will provide a written lay summary of the findings and also publish these on a study specific website, with contact information should they wish to discuss the findings. Our charity partners will be involved with feedback to the organisations they represent.

## CONCLUSION

At the time of writing (8 August 2019), a total of 736 participants have been randomised (380 to the intervention arm and 356 to the control arm), and the trial is now in the follow-up phase.

A comprehensive programme of preparatory work has allowed this substantial trial of an education and self-management intervention for people living with chronic headaches to be developed. There is a clear interest in taking part in the trial. Our follow-up will be complete in early 2020, and the final results will be published in 2021.

**Author affiliations**
[1]Warwick Medical School, Clinical Trials Unit, University of Warwick, Coventry, UK
[2]Warwick Medical School, University of Warwick, Coventry, UK
[3]Centre for Primary Care and Public Health, Queen Mary University of London, London, UK
[4]Warwick Medical School, Division of Health Sciences, University of Warwick, Coventry, UK
[5]Warwick Medical School, Warwick Evidence, University of Warwick, Coventry, UK
[6]Department of Psychology, Royal Holloway University of London, Egham, UK
[7]University College London Queen Square Institute of Neurology and The National Hospital for Neurology and Neurosurgery, London, UK

**Contributors** SP - study concept and design, wrote the first draft of the manuscript and finalised the manuscript for submission. FA, DC, SE, DE, FG, KH, SWH, DM, HM, VN, SPe, TP, RP, HS, SJT, KW, MSM - study concept and design, provided critical revisions to the manuscript. MU – conceived the original study design and supported writing of the first draft of the manuscript. DC, SE, DE, FG, KH, SWH, HM, SPe, TP, HS, SJT, MU, MSM - original grant holders.

**Funding** This research was funded by the NIHR Programme Grants for Applied Research programme (RP-PG- 1212-20018).

**Disclaimer** The views expressed in this publication are those of the author(s) and not necessarily those of the National Health Service, the National Institute for Health Research (NIHR) or the Department of Health.

**Competing interests** MU is a director and shareholder of Clinvivo Ltd. Use of this company was specified in original application for funding to NIHR. MU has recused himself from all discussion regarding the use of the app in this study. All contracting processes have been in accord with University of Warwick financial regulations. MU was chair of the National Institute for Health and Care Excellence accreditation advisory committee until March 2017 for which he received a fee. He is chief investigator or coinvestigator on multiple previous and current research grants from the UK NIHR, Arthritis Research UK and is a coinvestigator on grants funded by Arthritis Australia and Australian NHMRC. He has received travel expenses for speaking at conferences from the professional organisations hosting the conferences. He is part of an academic partnership with Serco Ltd related to return to work initiatives. He is an editor of the NIHR journal series and a member of the NIHR Journal Editors group for which he receives a fee. He has published multiple papers on chronic pain some of which are referenced in this paper. MM serves on the advisory board for Allergan, Medtronic, Novartis and TEVA and has received payment for the development of educational presentations from Allergan, electroCore, Medtronic, Novartis and TEVA. ShP and HKS are directors of Health Psychology Services Ltd, which in part provides psychological treatments for those with chronic pain.

**Patient consent for publication** Not required.

**Ethics approval** Ethics approval was given on the 17 February 2017 by North West – Greater Manchester East Research Ethics Committee (REC REF: 16/NW/0890). Written consent was taken for participation.

**Provenance and peer review** Not commissioned; externally peer reviewed.

**ORCID iDs**
Shilpa Patel http://orcid.org/0000-0003-0726-4888
Dawn Carnes http://orcid.org/0000-0002-3152-3133
David R Ellard http://orcid.org/0000-0002-2992-048X
Frances Griffiths http://orcid.org/0000-0002-4173-1438
Siew Wan Hee http://orcid.org/0000-0002-0415-263X
Hema Mistry http://orcid.org/0000-0002-5023-1160
Vivien P Nichols http://orcid.org/0000-0002-3372-1395
Stavros Petrou http://orcid.org/0000-0003-3121-6050
Rachel Potter http://orcid.org/0000-0001-6655-8996
Stephanie Taylor http://orcid.org/0000-0001-7454-6354
Martin Underwood http://orcid.org/0000-0002-0309-1708

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
