## [Reviewer comments · BMJ Open]

ARTICLE DETAILS

TITLE (PROVISIONAL)	Usual care and a self-management support programme vs usual care and a relaxation programme for people living chronic headache disorders: a randomised controlled trial protocol (CHESS)
AUTHORS	Patel, Shilpa; Carnes, Dawn; Eldridge, Sandra; Ellard, David R; Griffiths, Frances; Haywood, Kirstie; Hee, Siew Wan; Mistry, Dipesh; Mistry, Hema; Nichols, Vivien P; Petrou, Stavros; Pincus, Tamar; Potter, Rachel; Sandhu, Harbinder; Taylor, Stephanie; Underwood, Martin; White, Kimberley; Matharu, Manjit

VERSION 1 – REVIEW

REVIEWER	Francesca Puledda King's College London
REVIEW RETURNED	01-Oct-2019

GENERAL COMMENTS	Comments for authors: This is an interesting paper describing the methodology of a large randomized trial to determine the effectiveness and cost-effectiveness of a group self-management support programme for adults with chronic headaches, that is currently under way. The trial is well written and will certainly help to understand better the impact of chronic headache disorders in the general population. My main consideration lies in how the diagnosis of 'chronic headache' will be made. The primary contact patients have is through a telephone interview led by a nurse, based on a telephone questionnaire previously published by the same study group. It is not specified if the nurse conducting the interview will have been trained in headache disorders, or if the primary diagnosis underlying the chronic problem will be confirmed by a trained headache physician. As the authors rightly point out, the diagnosis of the underlying primary headache disorder is of paramount importance in defining the subsequent line of treatment. A minor point: In the Introduction, please clarify what is meant by 'chronic headaches'. ICHD definition distinguishes between chronic migraine, chronic tension type headache and NDPH. There is no such definition as 'chronic headache'. Also please be more specific about the definition of MOH: 'Headache occurring on 15 or more days/month in a patient with a pre-existing primary headache
--

	and developing as a consequence of regular overuse of acute or symptomatic headache medication, on 10 or more or 15 or more days/month, depending on the medication, for more than 3 months.'
--	---

REVIEWER	Aynur Özge Mersin University School of Medicine, Mersin, Turkey
REVIEW RETURNED	22-Oct-2019

GENERAL COMMENTS	It is a well written comprehensive methodological study It can be acceptable for publication
---

REVIEWER	Elida Zairina Faculty of Pharmacy, Universitas Airlangga, Surabaya, Indonesia
REVIEW RETURNED	05-Nov-2019

GENERAL COMMENTS	Dear Editor, Thank you for giving me the opportunity to review the manuscript entitled: Usual care and a self-management support programme vs usual care and a relaxation programme for people living chronic headache disorders: a randomised controlled trial authored by Dr Patel et al. protocol (CHES). This trial is promising a good result that will add an important aspect in the healthcare services. Please find below is my comments for the manuscript. GENERAL OVERVIEW I am not sure what the guideline is for authors in the BMJ OPEN regarding a research protocol manuscript. However, it will be good if this research protocol has a subheading about discussion that consist of the stage are this trial currently going? Are the recruitment is started already? How many participants? What are the expectation of this trial i.e. if this trial works etc. ABSTRACT: Again, there is no explanation about this trial, is the recruitment process is started already? If it is, how many participants that has been joined to the trial? INTRODUCTION: What are the gaps in the literature? Why this trial is important to conduct? Are there any similar trials? Were they works? More background about previous similar trial is advised. The authors include smartphone in the trial, should mentioned in the background if trial that involved smartphone app is actually works for self-management in chronic diseases. METHODS AND AIMS Page 8 line 22 – Table 1 (inclusion and exclusion) criteria: The inclusion and exclusion h should be explained clearly, since chronic headache the outcomes is sometimes subjective, measuring using objective tools is a bit difficult. I.e. Participants included if they aged 18 years and above: My question is: Are there any maximum age to include? For elderly with dementia or people with other co-morbidities such as chronic diseases, will they be included? Since the exclusion are those who has an underlying serious psychological disorder etc. Are those who has not smartphone will be excluded? This trial clearly will exclude those who has no access to a telephone, how about those who has not smartphone? Will they be excluded too?
---

	Having smartphone / or telephone – will it be possible to compare and measure the cost effectiveness as the outcome of the study? Since no telephone access is excluded. Page 9 line 11: The definition of the facilitators, what does it mean by health care professionals: pharmacist? Doctors? If at least a registered nurse as one of the facilitators in each group, how do the research team keep the balance of the intervention delivery if the group has different healthcare professionals? Although the protocol has mentioned about the training and monitoring for the facilitators. Page 9 line 40: It is unclear whether the participants in the intervention need to complete a paper headache diary or a smartphone app? Or both? Since in page 16 table 3, it shows about questions in smartphone app. It should be mentioned clearly what kind of activities that the intervention group need to complete (in page 9) Page 12 line 24 and page 16 Table 3. Are there any particular type of smartphone to download the app? IOS or android base? Will this app be available to public after the trial completed or if it proven to be effective for self-management in chronic headache? These kind of question should be mentioned in the discussion section to give such an interest topic to audience and curiosity to wait for the study result For the intervention group, will they get any reminder to fill their paper headache diary or smartphone app? Or a compliance in filling the data will be measured as an outcome whether this trial works or not? It should be clearly defined in the protocol. Page 15 line 31 I understand that is not possible to blind the participants, facilitators, or the central coordinating CHES team. But the research team should try to minimise bias as much as possible. It is unclear, whether there any outcome assessor or statistician blinding for collecting and analyse the data. Adding a Discussion section is advisable to see the expectation of this trial.
--	---

REVIEWER	Yohannes W Woldeamanuel Stanford University, USA Advanced Clinical & Research Center, Ethiopia Propria Health Solutions, USA
REVIEW RETURNED	08-Nov-2019

GENERAL COMMENTS	This is a well-designed research protocol for determining the efficacy of self-management program developed to treat chronic headache disorders. Self-management approaches are known to have lasting impact in chronic headache therapy; these approaches also increase internal locus of control, self-efficacy, and make patients proactive participants in their health management. I have minor comments as below. - any plan for baseline period pre-randomization? A 12-week baseline period is recommended by the IHS chronic migraine guidelines for clinical trials. This is advised to avoid regression to mean in chronic migraine prevention trials, and also to help assess and/or enforce participants' compliance in reporting outcome data.
---

	 - why is the control group slightly lower in sample size compared to treatment arm (self-management)? A minimum of 1:1 ratio is the recommendation for controlling effects. - is the study also designed to allow superiority or inferiority comparison trial design between self-management and usual care plus relaxation therapy? - any plan or thoughts to include crossover design? - Not clear to me as to why chronic migraine and chronic tension-type headache were mingled? Although, analysis will be stratified as described. - why was HIT-6 selected over MIDAS? HIT-6 is known to be more influenced by headache intensity while MIDAS by headache frequency. My suggestion is that it is better to combine both tools to obtain an overall and more accurate assessment of headache-related disability. - strength: multiple headache-related outcomes reported as per IHS guidelines to get overall headache improvement picture, inclusion of CACE analysis (as some patients may not be compliant in such long trials). - the authors have clarified the potential of clustering post-randomization with some ICC. I assume this study fits to the individually randomized group trial design. Any attempt to match confounders a priori? - how about utilizing the headache self-efficacy scale specifically designed for headache compared to PSEQ; or use both and compare results - bodily pain: how about PHQ-15?
--	---

VERSION 1 – AUTHOR RESPONSE

Reviewer: 1

My main consideration lies in how the diagnosis of ‘chronic headache’ will be made. The primary contact patients have is through a telephone interview led by a nurse, based on a telephone questionnaire previously published by the same study group. It is not specified if the nurse conducting the interview will have been trained in headache disorders, or if the primary diagnosis underlying the chronic problem will be confirmed by a trained headache physician. As the authors rightly point out, the diagnosis of the underlying primary headache disorder is of paramount importance in defining the subsequent line of treatment.

The diagnosis of chronic headache was made during an eligibility phone call with a member of research team who confirmed that the potential participant had had a headache for more than 15 days per month for the last three months. In the second stage the nurses were diagnosing the underlying headache disorder. The nurses conducting the telephone classification have received a days training delivered by a neurologist and senior researcher in charge of developing the tool. Our validation work for the tool shows very good reliability between classification made by a nurse and that by a headache specialist doctor.[1]

In the Introduction, please clarify what is meant by ‘chronic headaches’. ICHD definition distinguishes between chronic migraine, chronic tension type headache and NDPH. There is no such definition as

'chronic headache'. Also please be more specific about the definition of MOH: 'Headache occurring on 15 or more days/month in a patient with a pre-existing primary headache and developing as a consequence of regular overuse of acute or symptomatic headache medication, on 10 or more or 15 or more days/month, depending on the medication, for more than 3 months.'

We understand this reviewers concerns about being clear on our population of interest. Our view, however that there is a distinction between acknowledging the presence of chronic headache symptoms and providing a diagnosis of the underlying headache disorder. In this introduction for the non-specialist reader we are simply setting the scene for the magnitude of the problem. Similarly in the introduction we are introducing the general reader to the concept of medication overuse headaches. We have addressed in detail, in a previous paper, exactly how we operationalised ICHD criteria to define the population included on this trial.[2]

Reviewer: 2

We thank the reviewer for their interest in our paper and the positive comments provided.

Reviewer: 3

It will be good if this research protocol has a subheading about discussion that consist of the stage are this trial currently going? Are the recruitment is started already? How many participants? What are the expectation of this trial i.e. if this trial works etc.

We have signposted this more clearly in the paper.

ABSTRACT: Again, there is no explanation about this trial, is the recruitment process is started already? If it is, how many participants that has been joined to the trial?

As this is a protocol paper we have followed the guidance by BMJ open on the subheadings in the abstract. It is difficult to include details of the trial results here and therefore we have a paragraph at the end under the subheading, conclusion.

INTRODUCTION: What are the gaps in the literature? Why this trial is important to conduct? Are there any similar trials? Were they works? More background about previous similar trial is advised. The authors include smartphone in the trial, should mentioned in the background if trial that involved smartphone app is actually works for self-management in chronic diseases.

Within the strict word limit of BMJ proposal it is difficult to go into the literature for all aspects of the trial. We have presented here a brief summary of the rationale for this trial. We have summarised elsewhere the background literature and refer to these papers in our protocol.[1-8] The reviewer comments upon our use of a smartphone app. This is being used for data collection - not as part of our intervention and is not therefore described in the background. We have summarised its performance in our feasibility study elsewhere.[2] In a future paper we will describe its performance within the main trial.

METHODS AND AIMS:

Page 8 line 22 – Table 1 (inclusion and exclusion) criteria: The inclusion and exclusion h should be explained clearly, since chronic headache the outcomes is sometimes subjective, measuring using objective tools is a bit difficult. I.e. Participants included if they aged 18 years and above:

My question is: Are there any maximum age to include? For elderly with dementia or people with other co-morbidities such as chronic diseases, will they be included? Since the exclusion are those who has an underlying serious psychological disorder etc.

There was no maximum age limit, we have clarified that in table 1. In our recruitment sub-section we have detailed that participants were identified from screening of GP practices, the identified patient list was then screened by a GP in the practice whereby they were asked to flag anyone that might not be suitable. Here is where we would expect those with terminal illnesses, poorly controlled serious mental illness, dementia or other conditions that might make them unsuitable to be flagged. We did not exclude anyone with concurrent disorders so long as they were able to attend the group sessions and they did not have any serious psychological disorder that would preclude participation in the group sessions.

Are those who has not smartphone will be excluded? This trial clearly will exclude those who has no access to a telephone, how about those who has not smartphone? Will they be excluded too? Having smartphone / or telephone – will it be possible to compare and measure the cost effectiveness as the outcome of the study? Since no telephone access is excluded.

The access to a phone referred to in the inclusion and exclusion table is in reference to a phone in general and not specifically a smart phone (we have clarified that in table 1). We needed to be able to contact participants via phone to do the headache classification and therefore without this it would not be possible. For clarification the use of the smartphone was for data collection on headache frequency, severity and duration. Anyone without a smartphone was provided with a paper alternative.

Page 9 line 11:

The definition of the facilitators, what does it mean by health care professionals: pharmacist? Doctors? If at least a registered nurse as one of the facilitators in each group, how do the research team keep the balance of the intervention delivery if the group has different healthcare professionals? Although the protocol has mentioned about the training and monitoring for the facilitators.

We have clarified the health care professionals in this section. With regards to managing the balance, all the facilitators received the same training, with the nurses receiving the additional training on medication and classification as that was their remit to deliver. Facilitators agreed before the delivery which sections they would deliver and they were encouraged to split the delivery equally. All facilitators were observed as part of quality assessment and feedback was provided to each of them.

Page 9 line 40:

It is unclear whether the participants in the intervention need to complete a paper headache diary or a smartphone app? Or both? Since in page 16 table 3, it shows about questions in smartphone app. It should be mentioned clearly what kind of activities that the intervention group need to complete (in page 9) Page 12 line 24 and page 16 Table 3.

Participants in both arms of the study were asked to complete the smart phone app to record outcome. Only those randomised to the intervention arm were requested to complete the paper headache diary. We have clarified the paper diary on page 9. On page 12 all the outcome measures stated are completed by both groups the only additional measure completed by the intervention group was the paper diary which we have clarified on page 9. Page 16 table 3 represents the questions presented in the app, on page 15 line 45 we state all participants complete this.

Are there any particular type of smartphone to download the app? IOS or android base? Will this app be available to public after the trial completed or if it proven to be effective for self-management in chronic headache? These kind of question should be mentioned in the discussion section to give such an interest topic to audience and curiosity to wait for the study result.

Our app is available on both IOS and android and there was no restriction on the type of smartphone needed. To clarify the smartphone app is for data collection, it is not part of the intervention. A future paper is planned on its performance within the trial

For the intervention group, will they get any reminder to fill their paper headache diary or smartphone app? Or a compliance in filling the data will be measured as an outcome whether this trial works or not? It should be clearly defined in the protocol.

There are no specific reminders to complete the paper headache diaries by the intervention group. We do have a process of reminders for the smart phone app which we have included in the data collection methods section. The data collected in the smartphone app is not part of the intervention.

Page 15 line 31

I understand that is not possible to blind the participants, facilitators, or the central coordinating CHES team. But the research team should try to minimise bias as much as possible. It is unclear, whether there any outcome assessor or statistician blinding for collecting and analyse the data. Adding a Discussion section is advisable to see the expectation of this trial.

Our main outcomes are all from participant self-report. All data entry and checking is done blind to treatment allocation. The unbalanced randomisation means it is not

possible to blind the statisticians to randomisation. We have added some additional detail on blinding.

Reviewer: 4

Any plan for baseline period pre-randomization? A 12-week baseline period is recommended by the IHS chronic migraine guidelines for clinical trials. This is advised to avoid regression to mean in chronic migraine prevention trials, and also to help assess and/or enforce participants' compliance in reporting outcome data.

We are not using a baseline run-in period for this pragmatic trial. The point at which this intervention would fit into the care pathway is following a referral to the programme from their general practitioner. Replicating that as far as possible within the selection criteria for this trial will mean our results are directly applicable to clinical care. This would not be the case if we used a prolonged run-in period. We have clarified this in the text under eligibility criteria.

Why is the control group slightly lower in sample size compared to treatment arm (self-management)? A minimum of 1:1 ratio is the recommendation for controlling effects.

The sample size was calculated using a formula that accounts for grouping in one of the arms hence there are slightly fewer participants in the control arm. This is the most efficient design

Is the study also designed to allow superiority or inferiority comparison trial design between self-management and usual care plus relaxation therapy?

It is a superiority study.

Any plan or thoughts to include crossover design?

This a parallel group trial. A cross-over design would not be suitable for assessing the effect of a behavioural intervention.

Not clear to me as to why chronic migraine and chronic tension-type headache were mingled? Although, analysis will be stratified as described.

This is a pragmatic trial. The point at which this intervention would fit into the care pathway is following a referral to the programme from their general practitioner. At this point headache type will not have been classified. The provision of advice on likely diagnosis is part of the intervention. We have added some text to explain this point under eligibility criteria.

Why was HIT-6 selected over MIDAS? HIT-6 is known to be more influenced by headache intensity while MIDAS by headache frequency. My suggestion is that it is better to combine both tools to obtain an overall and more accurate assessment of headache-related disability.

In our review of outcome measures (Haywood et al, Cephalalgia, 2018, Vol. 38(7) 1374–1386) we assessed 23 PROMS including the HIT-6 and MIDAS and concluded only three measure had acceptable evidence of reliability and validity. The HIT-6 was the only measure that had acceptable evidence supporting its completion by all headache populations. As our trial focused on chronic headache and not only migraine this was deemed a good measure.

The authors have clarified the potential of clustering post-randomization with some ICC. I assume this study fits to the individually randomized group trial design. Any attempt to match confounders a priori?

The unit of analysis is the individual. Randomisation is stratification by locality and headache type. We will include these and other covariates in our final analysis as described in our statistical analysis plan.

How about utilizing the headache self-efficacy scale specifically designed for headache compared to PSEQ; or use both and compare results

Thank you for your helpful suggestion, as we are now in follow-up it this is not something we can amend but it may be a point of discussion once we know the final results.

Bodily pain: how about PHQ-15?

Thank you for the suggestion. As the trial is now in follow-up phase this maybe something we can reflect on once we know the results.

References

1. Potter R, Hee SW, Griffiths F, et al. Development and validation of a telephone classification interview for common chronic headache disorders. *J Headache Pain* 2019;**20**(1):2.
2. White K, Potter R, Patel S, et al. Chronic Headache Education and Self-management Study (CHES) - a mixed method feasibility study to inform the design of a randomised controlled trial. Submitted. 2019;**19**(30).
3. Probyn K, Bowers H, Caldwell F, et al. Prognostic factors for chronic headache: A systematic review. *Neurology* 2017;**89**(3):291-301.
4. Nichols VP, Ellard DR, Griffiths FE, Kamal A, Underwood M, Taylor SJC. The lived experience of chronic headache: a systematic review and synthesis of the qualitative literature. *BMJ open* 2017;**7**(12):e019929.
5. Probyn K, Bowers H, Mistry D, et al. Non-pharmacological self-management for people living with migraine or tension-type headache: a systematic review including analysis of intervention components. *BMJ open* 2017;**7**(8):e016670.
6. Patel S, Potter R, Matharu M, et al. Development of an education and self-management intervention for chronic headache - CHES trial (Chronic Headache Education and Self-management Study). *J Headache Pain* 2019;**20**(1):28.
7. Haywood KL, Mars TS, Potter R, Patel S, Matharu M, Underwood M. Assessing the impact of headaches and the outcomes of treatment: A systematic review of patient-reported outcome measures (PROMs). *Cephalalgia : an international journal of headache* 2018;**38**(7):1374-86.
8. Nichols VP, Ellard DR, Griffiths FE, Underwood M, Taylor SJC, Patel S. The CHES trial: protocol for the process evaluation of a randomised trial of an education and self-management intervention for people with chronic headache. *Trials* 2019;**20**(1):323.

VERSION 2 – REVIEW

REVIEWER	Francesca Puledda King's College London
REVIEW RETURNED	09-Dec-2019

GENERAL COMMENTS	The authors seem to have addressed the reviewer's concerns.
---

REVIEWER	Elida Zairina Faculty of Pharmacy, Universitas Airlangga
REVIEW RETURNED	29-Nov-2019

GENERAL COMMENTS	Thank you for making the amendments and for addressing my queries and suggestions. The authors have done a
--

	comprehensive job in addressing all my comments and I do not have any other queries or further suggestions. I believe this is a good piece of work and the authors have done well in undertaking the research work and in producing this comprehensive manuscript. I look forward for the results of the study when its published
REVIEWER	Yohannes W Woldeamanuel Division of Headache Stanford University School of Medicine CA, USA Advanced Clinical & Research Center Addis Abeba, Ethiopia Propria Health Solutions CA, USA
REVIEW RETURNED	03-Dec-2019
GENERAL COMMENTS	- I do NOT accept the term "chronic headache", it must be removed from the manuscript. The reference #3 does not define "chronic headache". There is no diagnosis or condition known as "chronic headache". Please rephrase it to indicate what you are studying - chronic migraine (CM) and chronic tension-type headache (CTTH). And avoid using the term "chronic headache". - Please rephrase the title to "pragmatic controlled trial" instead of "randomized controlled trial". This will help justify most of the issues typical of RCT that are missed in this protocol. - I suggest including more multi-center/heterogenous settings for future: effectiveness studied in this PCT

VERSION 2 – AUTHOR RESPONSE

Reviewer: 1

We thank the reviewer for their interest in our paper and the positive comments provided.

Reviewer: 3

We thank the reviewer for taking the time to review our paper.

Reviewer: 4

I do NOT accept the term "chronic headache", it must be removed from the manuscript. The reference #3 does not define "chronic headache". There is no diagnosis or condition known as "chronic headache". Please rephrase it to indicate what you are studying - chronic migraine (CM) and chronic tension-type headache (CTTH). And avoid using the term "chronic headache".

The reviewer is correct that there is not a diagnosis of chronic headache. However, our study is not of people with a diagnosed headache disorder. Being provided with a classification of their likely headache type is part of the intervention. If the CHES intervention is effective the point it will sit in the care pathway is for people with very frequent, but undiagnosed, headaches. Some terminology is needed to describe this group. The term 'chronic daily headache' is widely used in the literature but it is not recognised by HIS/ICHD and inconsistently defined in the literature. It is a misleading approach to use for our current study since it implies headaches on every day when this is not part of the diagnostic criteria for chronic migraine or chronic tension type headache. Our population of interest for this study are people with chronic migraine or chronic tension type headache, with or without medication overuse headache. For our chronic headache disorders of interest the definition of chronicity is 'headache for 15 or more days per month for at least three months'. We have, therefore

used this as our definition of chronicity. This general approach has been used in other recent epidemiological studies of chronic headache.[1-3]

Whilst a detailed discussion of these issues is beyond the scope of this current paper we have edited our introduction to clarify our approach to this.

1. Henning V, Katsarava Z, Obermann M, Moebus S, Schramm S. Remission of chronic headache: Rates, potential predictors and the role of medication, follow-up results of the German Headache Consortium (GHC) Study. *Cephalalgia*. 2018;38(3):551-560.
2. Kristoffersen ES, Lundqvist C, Russell MB. Illness perception in people with primary and secondary chronic headache in the general population. *J Psychosom Res*. 2019;116:83-92.
3. Westergaard ML, Lau CJ, Allesøe K, Gjendal ST, Jensen RH. Monitoring chronic headache and medication-overuse headache prevalence in Denmark. *Cephalalgia*. 2019 Sep 15.

Please rephrase the title to "pragmatic controlled trial" instead of "randomized controlled trial". This will help justify most of the issues typical of RCT that are missed in this protocol.

The study was funded as a randomised controlled trial by NIHR and therefore we feel strongly that this should remain as the title.

I suggest including more multi-centre/heterogeneous settings for future: effectiveness studied in this PCT.

We thank the reviewer for this suggestion.